# Total daily physical activity, brain pathologies, and parkinsonism in older adults

Shahram Oveisgharan[1,2]*, Robert J. Dawe[1,3], Sue E. Leurgans[1,2], Lei Yu[1,2], Julie A. Schneider[1,2,4], David A. Bennett[1,2], Aron S. Buchman[1,2]

1 Rush Alzheimer's Disease Center, Rush University Medical Center, Chicago, Illinois, United States of America, 2 Department of Neurological Sciences, Rush University Medical Center, Chicago, Illinois, United States of America, 3 Department of Radiology and Nuclear Medicine, Rush University Medical Center, Chicago, Illinois, United States of America, 4 Department of Pathology, Rush University Medical Center, Chicago, Illinois, United States of America

* shahram_oveisgharan@rush.edu

## Abstract

**Data Availability Statement:** All data included in these analyses are available via the Rush Alzheimer's Disease Center Research Resource Sharing Hub, which can be found at www.radc.

### Objective

We examined the association of physical activity, postmortem brain pathologies, and parkinsonism proximate to death in older adults.

### Methods

We studied the brains of 447 older decedents participating in a clinical-autopsy cohort study. We deployed a wrist worn activity monitor to record total daily physical activity during everyday living in the community-setting. Parkinsonism was assessed with 26 items of a modified motor portion of Unified Parkinson's Disease Rating Scale (UPDRS). We used linear regression models, controlling for age and sex, to examine the association of physical activity with parkinsonism with and without indices of Alzheimer's disease and related disorders (ADRD) pathologies. In separate models, we added interaction terms to examine if physical activity modified the associations of brain pathologies with parkinsonism.

### Results

Mean age at death was 90.9 (SD, 6.2), mean severity of parkinsonism was 14.1 (SD, 9.2, Range 0–59.4), and 350 (77%) had evidence of more than one ADRD pathologies. Higher total daily physical activity was associated with less severe parkinsonism (Estimate, -0.315, S.E., 0.052, p<0.001). The association of more physical activity with less severe parkinsonism persisted after adding terms for ten brain pathologies (Estimate, -0.283, S.E., 0.052, p<0.001). The associations of brain pathologies with more severe parkinsonism did not vary with the level of physical activity.

rush.edu. It has descriptions of the studies and available data. Any qualified investigator can create an account and submit requests for deidentified data.

**Funding:** This work was supported by National Institute of Health grants: A.S.B.: R01AG47679; R01AG056352; R01AG059732 D.A.B.: P30AG10161; R01AG017917; R01AG15819; R01AG043379 The funders had no role in study design, data collection and analysis, decision to publish, or preparation of the manuscript.

**Competing interests:** The authors have declared that no competing interests exist.

## Conclusion

The association of higher physical activity with less severe parkinsonism may be independent of the presence of ADRD brain pathologies. Further work is needed to identify mechanisms through which physical activity may maintain motor function in older adults.

## Introduction

Parkinsonism, a complex aging phenotype, includes impaired gait and balance, bradykinesia, rigidity, and tremor and may affect 50% or more of adults 85 years or older[1]. The presence of clinical parkinsonism is associated with an increased risk of disabilities[2], mild cognitive impairment[3], and dementia[4]. Given the magnitude of the personal and social consequences of parkinsonism, modifiable risk factors like physical activity are being intensely studied for their potential efficacy to maintain or reduce the severity of parkinsonism in older adults[5]. The neurobiology underlying the potential efficacy of a higher level of physical activity to reduce the severity of parkinsonism is unknown.

In a previous study, we found that a higher level of total daily physical activity in older adults is associated with less severe parkinsonism [6]. In several other studies, we have also shown that postmortem indices of brain pathology such as macroinfarcts are related to more severe parkinsonism[7,8], and to less total daily physical activity[9]. Together, these prior studies show that both physical activity and brain pathologies are related to the severity of parkinsonism. As illustrated in Fig 1, these reports provide the scientific framework and support for testing the hypothesis that brain pathologies link (mediates) physical activity with parkinsonism in older adults. If we do not find evidence for mediation, this would suggest that physical activity and brain pathologies are independently associated with parkinsonism in older adults. Thus, testing this hypothesis would provide novel data about a potential mechanism

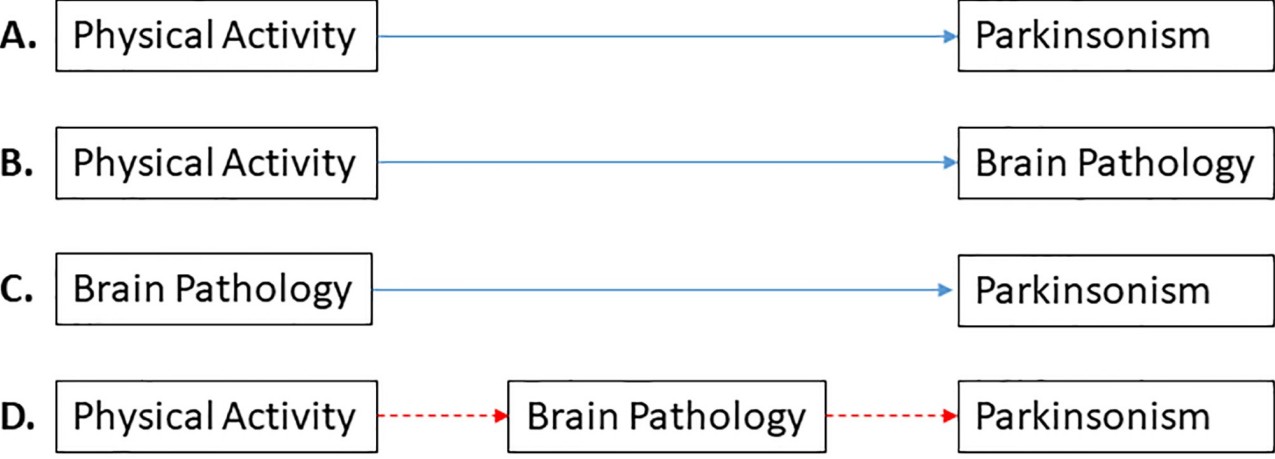

**Fig 1. Scientific framework for Hypothesis Testing.** Our prior work has shown that a higher level of physical activity is related to less severe parkinsonism (**A** [6]), physical activity and brain pathologies are related (**B** [9]), and brain pathologies are related to the severity of parkinsonism (**C** [7,8,10]). Based on these findings, this study tested the hypothesis that brain pathology links (mediates) the association of physical activity with parkinsonism. If the hypothesized sequence of events shown in **D** is correct, then adding the term for brain pathology to the model would attenuate the association of physical activity with parkinsonism, which would no longer be significant. If brain pathology does not mediate this association, both physical activity and brain pathology would be independently associated with parkinsonism.

underlying the motor benefits of physical activity in older adults. In turn this would provide novel targets for interventions to maintain motor function in old age.

Prior investigators did not test this mediation hypothesis since it is uncommon to obtain postmortem indices from large numbers of well-characterized older adults prior to death. To test the hypothesis that brain pathologies link physical activity with the severity of parkinsonism, we used novel data from 447 deceased participants from the Rush Memory and Aging Project[11].

## Methods

### Participants

The Rush Memory and Aging Project (MAP) is an ongoing cohort study of community-dwelling healthy older adults 65 years older or older who agree to annual clinical assessments and organ donation at the time of death. The study was approved by an Institutional Review Board of Rush University Medical Center. Written informed consent was obtained from all study participants as was an Anatomical Gift Act for organ donation. Participants are recruited from retirement facilities, subsidized housings, and individual homes across Chicago Metropolitan area. While the study's recruitment started in 1997, multi-day physical activity recordings started in 2005.

This study focuses on the associations between total daily physical activity metrics extracted from multiday activity recordings obtained in the community setting, brain indices of ADRD obtained at autopsy, and the severity of clinical parkinsonism based on the United Parkinson's Disease Rating Scale (UPDRS). As such, inclusion was limited to decedents with completed brain autopsy with valid measures of quantitative physical activity metrics and clinical parkinsonism assessment proximate to death. Since 2005, 613 participants with physical activity metrics died and 537 of them underwent autopsy (88% autopsy rate). Of these, 519 (97%) had completed autopsy results at the time of these analyses, 72 had missing clinical data (e.g. parkinsonism) which left 447 for these analyses. Comparison of participants included in the current analyses with participants excluded (S1 Table) showed that the two groups were not different in age, education, vascular risk factors and diseases, and levels of physical activity, but were different in the percentage of women and neuroleptic medication consumption (S1 Table).

### Assessment of parkinsonism

Trained nurse clinicians assessed parkinsonism annually using 26 items of a modified version of UPDRS[12,13]. This assessment has high inter-rater reliability and short-term stability among nurses and compared with a movement disorders specialist[12,13].

This testing assessed four common parkinsonian signs: parkinsonian gait, bradykinesia, rigidity, and tremor. Each of the 26 items was scored on a similar 0–4 scale, with 0 indicating no impairment and 4 indicating very severe impairment. We summarized scores for each of the four parkinsonian signs and the scores for these four signs were then averaged together to form a summary global parkinsonism score for the overall severity of parkinsonism as previously described. This measure was the primary outcome measure in our analyses[12,13].

### Assessment of total daily physical activity

Each participant wore an activity monitor (Actical; Philips Healthcare, Andover, MA) over the non-dominant hand 24 hours/day for up to 10 days. The Actical is an omni-directional accelerometer which generates a signal proportional to the magnitude of all movements it registers. The device digitizes the signal and expresses the average activity for each recording epoch as

activity counts (15s in this study; 5760 epochs/day). More physical activity is reflected in higher activity counts.

In prior work we constructed and validated two measures which summarize the mean total daily physical activity based on multiday recording[6,14]. In this study, we analyzed total daily activity for all days with complete 24 hours recordings. Total daily physical activity is the mean sum of all activity counts for up to 10 days of complete recordings. Since the benefits of physical activity may be related to the intensity of activity, we constructed a second measure. Intensity of daily physical activity was calculated by dividing the total daily activity counts by the total duration of non-zero 15 second epochs to yield a metric of the intensity of activity counts per hour of activity. These two metrics were validated in prior studies predicting adverse health outcomes and are independently associated with these outcomes when controlling for self-reported physical activity[6,14].

Movement during sleep and non-rest periods may make different contributions to parkinsonism. Sleep diaries are not available in this cohort and many participants nap extensively during the daytime making it difficult to filter out movements related to periods of sleep/rest. In prior studies, we developed and validated a sleep fragmentation metric $k_{RA}$, from actigraphic recordings [15]. $K_{RA}$ captures the temporal organization of human rest-activity patterns in terms of transition probabilities between periods of rest (when recorded activity is zero) to recording epochs in which movement (non-zero activity) is recorded. Conceptually, $k_{RA}$ is a measure of the tendency to fragment sustained rest periods by physical activity. A higher value of $k_{RA}$ represents more fragmented rest, and a lower value represents a more consolidated rest. Since most rest periods are recorded during sleep, a higher $k_{RA}$ is associated with more frequent movement during sleep. $K_{RA}$ is weakly related to total daily physical activity (Spearman correlation coefficient = 0.15, p = 0.003), suggesting that they measure different constructs.

These continuous multiday recordings were obtained during routine living in the community-setting and captured all movements including both exercise and habitual physical activity. While these inform on total levels of physical activity, the particular device employed cannot be used to quantify the contributions of specific activities. Thus, these metrics quantify total daily physical activity which shows how active an individual's lifestyle was during these multi-day recordings.

## Assessment of postmortem brain pathologies

Structured brain autopsies were performed by staff blinded to clinical data. Following brain removal, tissue preparation and sectioning, gross inspection, and tissue blocks were obtained from pre-specified brain regions as previously described[16].

**Alzheimer's disease pathology.** A modified Bielschowsky silver stain was used to visualize diffuse plaques, neuritic plaques, and neurofibrillary tangles, the AD pathological indices, in the frontal, temporal, parietal, entorhinal and hippocampal cortical areas[16]. Each of the three AD pathological indices was counted in each of the brain regions. For each AD pathological index, we created a summary measure by counting the index in each brain region and standardizing the count followed by making an average of the standardized scores across the 5 brain regions. Then, we constructed the global AD pathology score by averaging the summary measures of the three AD pathologic indices. We previously showed that each regional standardized score, e.g. neuritic plaques in the frontal cortex, was correlated with the other standardized AD indices' regional scores (r range: 0.47–0.73, median r = 0.67) and the Cronbach $\alpha$ coefficient was 0.90, indicating high internal consistency and supporting use of the global AD pathology score[17].

**Lewy body pathology.**   Immunohistochemistry with α-synuclein immunostain (Zymed; 1:50) was used for assessment of Lewy bodies on sections from the frontal, temporal, parietal, anterior cingulate, entorhinal, hippocampus, basal ganglia, and midbrain[16]. In this study, we used a dichotomous summary variable indicative of the presence or absence of Lewy bodies pathology.

**Nigral neuronal loss.**   Dissection of the diagnostic blocks included a hemisection of the midbrain containing subtantia nigra. Nigral neuronal loss was assessed in the substantia nigra at the level of the 3$^{rd}$ nerve exit using H&E staining. We used a semi-quantitative scale (none, mild to severe) for the assessment of nigral neuronal loss, as described previously[8].

**Transactive response DNA-binding protein 43 (TDP-43).**   TDP-43 was assessed using immunostaining with monoclonal antibody to phosphorylated TDP-43 (pS409/410; 1:100) in 6 brain regions: amygdala, hippocampus, dentate gyrus, entorhinal, frontal, and temporal cortices. Each of the 6 regions was assessed for the presence of the TDP-43 cytoplasmic inclusions in the glia or neurons, as described previously[16]. In this study, we used a dichotomous variable of the presence or absence of TDP-43 in the limbic or neocortical regions.

**Hippocampal Sclerosis (HS).**   HS was evaluated unilaterally in a coronal section of the midhippocampus at the level of lateral geniculate body, as described elsewhere[18]. It was graded as absent or present based on severe neuronal loss and gliosis in CA1 and/or subiculum.

**Macroinfarcts.**   Uniformed inspection for cerebral infarcts was conducted by naked eyes on the fixed slabs from one hemisphere and pictures of fresh slabs from the other hemisphere, and found lesions were confirmed microscopically[19]. For this analysis, we only included chronic infarcts, as acute or subacute infarcts were unlikely to affect parkinsonism measured on average two years prior to death. We used a dichotomous variable summarizing the presence or absence of the macroinfarcts.

**Microinfarcts.**   Microinfarcts are not visible to the naked eyes and can only be identified under microscopy, and have been shown to be associated with the parkinsonian signs[7]. A minimum of 9 regions in 1 hemisphere were examined for the microinfarcts: 6 cortical (frontal, temporal, entorhinal, hippocampal, parietal, and anterior cingulate), 2 subcortical (anterior basal ganglia and thalamus), and mid brain. Like macroinfarcts, only chronic microinfarcts were included for these analyses through a dichotomous variable (presence vs. absence of the microinfarcts).

**Atherosclerosis.**   Atherosclerosis severity was assessed by evaluation of circle of Willis vessels at the base of the brain (vertebral, basilar, posterior, middle, and anterior cerebral arteries). Severity of atherosclerosis was scaled semi-quantitatively (none, mild to severe) on the basis of atherosclerosis severity in each artery and number of affected arteries, as described previously [16]. For this study, we constructed a dichotomous variable for the presence or absence of moderate to severe atherosclerosis.

**Arteriolosclerosis.**   Arteriosclerosis severity was assessed by evaluation of the vessel walls in small arterioles of the anterior basal ganglia. It was based on the concentric hyaline thickening and narrowing of the examined vessels, and was scored semi-quantitatively (none, mild to severe, complete occlusion)[16]. For this study, we constructed a dichotomous variable for the presence or absence of moderate to severe arteriolosclerosis.

**Cerebral Amyloid Angiopathy (CAA).**   CAA was examined using amyloid-β immunostaining of meningeal and parenchymal vessels in four regions (frontal, temporal, angular, and calcarine cortices). CAA was scaled semi-quantitatively in each brain region (from 0 to 4) and CAA score was the average of the regional scores. From this continuous CAA score, we have also constructed a semi-quantitative (none, mild to severe) CAA scale[16]. For this analysis, we constructed a dichotomous variable for the presence or absence of moderate to severe CAA.

## Other clinical covariates

Sex and years of education were recorded at study entry. Age was calculated from date of birth to date of physical activity assessment. Chronic health conditions included the sum of three self-reported vascular risk factors (hypertension, diabetes mellitus, and smoking) and sum of four self-reported vascular diseases (myocardial infarction, congestive heart failure, stroke, and claudication). Self-reported marital status was assessed and classified into five categories: never married, married, widowed, divorced, and separated. All clinical data used in these analyses were obtained from the same visit proximate to death.

As done in prior studies, a diagnosis of Parkinson disease (PD) was based on self-reported medical history of a clinical diagnosis of PD for which the participants had received levodopa or dopamine agonists[20]. At each visit, study personnel reviewed and recorded all medications including neuroleptics taken by the study participants.

## Statistical analysis

The global parkinsonism scores were positively skewed, and these scores were square root transformed for these analyses [6,20]. As illustrated in Fig 1, we employed a series of multivariable linear regressions, adjusted for age at death and sex, to assess whether indices of brain pathologies link (mediate) the association of physical activity with the severity of parkinsonism proximate to death (Fig 1 and 1D). First, in a model without postmortem indices, we replicated the finding that physical activity was related to the severity of parkinsonism (Fig 1 and 1A). Second, we examined the association of total daily physical activity with ADRD pathology indices (Fig 1 and 1B). Third, we examined the association of postmortem indices with the severity of parkinsonism in a model without total daily physical activity (Fig 1 and 1C). Fourth, we examined a model that included both physical activity and brain pathologies together (Fig 1 and 1D). If the association of physical activity with parkinsonism is attenuated in model D, it would suggest that brain pathologies may link (mediate) the association of total daily physical activity with the severity of parkinsonism. In contrast, if the association of physical activity with the severity of parkinsonism does not change in the presence of brain pathology indices it will suggest that physical activity and brain pathologies are independently associated with the severity of parkinsonism. To examine potential collinearity among pathology indices, we calculated their variance inflation factor (VIF). VIF greater than 10 strongly suggests collinearity [21]. Since other health conditions and covariates may affect parkinsonism and physical activity, in further analyses we controlled for possible confounding effects of education, marital status, vascular risk factors and diseases, and use of neuroleptic medications. Finally, we repeated the models after exclusion of decedents with a clinical diagnosis of PD or decedents receiving neuroleptic medications that can affect the severity of parkinsonism.

Next, we repeated these analyses, but replaced total daily physical activity with the intensity of total daily physical activity, a second metric that captures aspects of daily physical activity that may be important for the benefit of physical activity. As the association of parkinsonism with sleep related movement may be different compared to movement during wake periods, we repeated the analyses by further adjustment for $k_{RA}$ [15], which is the probability of sleep disruption by movement.

Multiple mechanisms may account for the motor benefits of higher levels of physical activity. Prior studies have reported that life style factors are associated with less cognitive and motor impairment through modifying association of ADRD pathologies with cognitive and motor outcomes [22–25]. Therefore, in secondary analyses, we added interaction terms between physical activity and brain pathologies in model D to determine if the motor benefit

of physical activity was due to modification of the association between brain pathologies and parkinsonism. The analyses were done using SAS version 9.4.

# Results

## Characteristics of study participants

*Clinical Characteristics*: There were 447 participants and their clinical and postmortem indices are summarized in Table 1. On average total daily physical activity was measured 2.1 (SD = 2.0) years prior to death, and the average length of recording was for 9.4 (SD = 1.8) days.

**Postmortem indices.** The average postmortem interval was 9.0 (SD = 7.7) hours. For descriptive purposes we dichotomized the presence or absence of each of the 10 indices of brain pathologies as summarized in Table 1. The average participant showed evidence of 3 different pathologies (SD = 1.7, median = 3).

## Total daily physical activity, brain pathologies, and parkinsonism

First, we replicated prior findings that a higher level of total daily physical activity proximate to death was associated with less severe parkinsonism (Fig 1A; Table 2, **Model 1-TDPA**). Second, we replicated prior findings that a higher level of physical activity was associated with lower odds of nigral neuronal loss and macroinfarcts (Fig 1B; S2 Table).

Third, we found that a higher burden of nigral neuronal loss and atherosclerosis were associated with more severe parkinsonism proximate to death (Fig 1C; Table 2, **Model 2-Path**). The VIFs in this model were less than 1.5 indicating that collinearity did not exist among the examined pathology indices [21].

Fourth, we examined the association of total physical activity with the severity of parkinsonism in a model which also included terms for brain pathologies. In this model, the association of total daily physical activity with parkinsonism was not attenuated by the presence of brain pathologies (Fig 1D; Table 2, **Model 3-TDPA+Path**). In the **Model 3-TDPA-Path** model, total daily physical activity was independently associated with parkinsonism accounting for an additional 5.7% of the variance of parkinsonism as compared to 8.3% contributed by age, sex and brain pathologies.

In a series of sensitivity analyses, we examined the association of total daily physical activity with parkinsonism severity in models adjusting for education, marital status, vascular risk factors and diseases, and consumption of neuroleptic medications as possible confounding variables. Adjustment for these covariates did not affect the finding that higher levels of total daily physical activity was associated with less severe parkinsonism independent of brain pathology indices (S3 Table). Participants with a clinical diagnosis of PD (N = 17) or those taking neuroleptics (N = 86) might have more severe parkinsonism. After excluding these participants, our results were unchanged (S4 Table).

## Other physical activity metrics, brain pathologies, and parkinsonism

In a series of similar models, we replaced total daily physical activity with the intensity of daily activity. Controlled for age and sex, more intense physical activity was associated with less severe parkinsonism (estimate = -4.406, SE 0.671, p < 0.001), and the association was unaffected by the presence of the terms of brain pathologies (Table 2, **Model 4-Intensity+Path**).

The association of parkinsonism with sleep related movement may be different compared to movement during wake periods. To quantify sleep-related movement, we used $k_{RA}$ [15], which is the probability of sleep disruption by movement. We examined if adding a term for

**Table 1. Clinical and postmortem measures of the participants in these analyses.**

| Covariates | Summary measure |
|---|---|
| **Demographic** | **Mean (SD) or N (%)** |
| Age at death (years) | 90.9 (6.2) |
| Age at the actigraphic recording | 88.8 (6.2) |
| Female | 316 (71) |
| Years of education | 14.7 (2.9) |
| Marital status | |
| Never married | 34(8) |
| Married | 97(23) |
| Widowed | 261(62) |
| Divorced | 26(6) |
| Separated | 0 |
| **Clinical** | |
| Sum of history of vascular risk factors | 1.3 (0.8) |
| Hypertension | 315 (70) |
| Diabetes Mellitus | 97 (22) |
| Smoking (ever in life) | 176 (40) |
| Sum of vascular diseases | 0.8 (0.9) |
| Stroke | 95 (22) |
| Heart attack | 87 (19) |
| Heart failure | 57 (15) |
| Lower extremities claudication | 121 (27) |
| Neuroleptic medication use | 86 (19) |
| *Severity of parkinsonism* | **Mean (range)** |
| Global parkinsonism score | 14.1 (0–59.4) |
| Parkinsonian gait score | 33.5 (0–100) |
| Bradykinesia score | 17.2 (0–80) |
| Rigidity score | 3.9 (0–60) |
| Tremor score | 3.1 (0–45.5) |
| *Quantitative physical activity metrics* | **Mean (SD)** |
| Total Daily physical activity (activity counts/day) | $1.44 \times 10^5$ $(1.13 \times 10^5)$ |
| Intensity of physical activity (activity counts/active hours) | $0.18 \times 10^5$ $(0.09 \times 10^5)$ |
| $K_{RA}$ (an indirect measure of sleep time movement) | 0.028 (0.008)* |
| **Postmortem Indices** | **N (%)** |
| *Neurodegenerative* | |
| NIA-Reagan AD pathological diagnosis | 309 (69) |
| Nigral neuronal loss (moderate or severe) | 49 (11) |
| Lewy bodies (present in one or more sites) | 125 (28) |
| Limbic or neocortical Transactive response DNA-binding protein-43 (TDP-43) | 159 (36) |
| Presence of hippocampal sclerosis | 47 (11) |
| *Cerebrovascular pathologies* | |
| Macroinfarcts (1 or more present) | 175 (39) |
| Microinfarcts (1 or more present) | 144 (32) |
| Atherosclerosis (moderate or severe) | 116 (26) |
| Arteriolosclerosis (moderate or severe) | 134 (30) |
| Cerebral Amyloid Angiopathy (moderate or severe) | 149 (33) |
| Number of pathologies present | 3.1 (1.7) |

*$K_{RA}$ was available for 378 of the participants.

**Table 2. Total daily physical activity, indices of brain pathologies and parkinsonism proximate to death[*].**

| Model Terms | Model 1-TDPA | Model 2-Path | | Model 3-TDPA+Path | Model 4-Intensity+Path |
| --- | --- | --- | --- | --- | --- |
| | Est. (SE) | Est. (SE) | Variance inflation factor | Est. (SE) | Est. (SE) |
| | p-value | p-value | | p value | p-value |
| **Total daily physical activity (TDPA)** | -0.315 (0.052) | | | -0.283 (0.052) | |
| | <0.001 | | | <0.001 | |
| **$K_{RA}$** | | | | | |
| **Intensity of total daily physical activity (intensity)** | | | | | -4.022 (0.669) |
| | | | | | <0.001 |
| **AD pathology** | | -0.157 (0.105) | 1.2 | -0.151 (0.102) | -0.144 (0.101) |
| | | 0.135 | | 0.138 | 0.153 |
| **Lewy body pathology** | | 0.057 (0.143) | 1.2 | 0.029 (0.138) | 0.006 (0.137) |
| | | 0.689 | | 0.834 | 0.964 |
| **Nigral neuronal loss** | | 0.558 (0.203) | 1.2 | 0.472 (0.197) | 0.500 (0.195) |
| | | 0.006 | | 0.017 | 0.011 |
| **TDP-43** | | -0.124 (0.135) | 1.3 | -0.058 (0.131) | -0.050 (0.130) |
| | | 0.357 | | 0.659 | 0.701 |
| **Hippocampal sclerosis** | | 0.256 (0.206) | 1.2 | 0.210 (0.200) | 0.232 (0.198) |
| | | 0.215 | | 0.294 | 0.243 |
| **Macroinfarcts** | | 0.233 (0.125) | 1.1 | 0.177 (0.122) | 0.196 (0.121) |
| | | 0.063 | | 0.147 | 0.106 |
| **Microinfarcts** | | 0.025 (0.128) | 1.1 | 0.012 (0.124) | -0.009 (0.123) |
| | | 0.843 | | 0.925 | 0.939 |
| **Arteriolosclerosis** | | 0.196 (0.133) | 1.1 | 0.195 (0.128) | 0.154 (0.128) |
| | | 0.140 | | 0.130 | 0.229 |
| **Atherosclerosis** | | 0.376 (0.140) | 1.1 | 0.380 (0.135) | 0.388 (0.134) |
| | | 0.007 | | 0.005 | 0.004 |
| **Cerebral Amyloid Angiopathy** | | 0.005 (0.129) | 1.1 | 0.034 (0.125) | 0.027 (0.124) |
| | | 0.970 | | 0.784 | 0.829 |
| **Additional Explained Variance[**]** | 7.2% | 5.8% | | 11.5% | 12.7% |

[*]All the models are linear regressions and included terms for age and sex which alone accounted for 2.5% of the variance of the outcome which is square root transformation of the global parkinsonism score. Cells' parameters are estimates (SE, p value) derived from the linear regressions.

[**]In each model, an additional 2.5% of the variance is due age at death and sex.

$k_{RA}$ to model **A** confounded the association between total daily physical activity and parkinsonism. In this model, a higher level of total daily physical activity remained associated with less severe parkinsonism, while a higher level of $k_{RA}$ was associated with more severe parkinsonism (S5 Table). These finding suggest that a higher level of physical activity is associated with less severe parkinsonism and increased sleep fragmentation, due to more frequent movement during sleep, is associated with more severe parkinsonism. These associations persisted in the presence of brain pathologies (S5 Table).

## Total daily physical activity and modification of the association between brain pathologies and parkinsonism

Physical activity might also be associated with less severe parkinsonism through a second mechanism by modifying the untoward contribution of brain pathologies to the severity of parkinsonism. Therefore, we examined if there was evidence to support this mechanism by addition of interaction terms for total daily physical activity and brain pathologies to Model D

(Fig 1). The association of the pathology indices with the parkinsonism did not vary with the level of total daily physical activity (nigral neuronal loss × total daily physical activity [estimate = -0.323, SE 0.233, p = 0.166]; and atherosclerosis × total daily physical activity [estimate = -0.216, SE 0.114, p = 0.060]). These additional analyses suggest that the association of a higher level of physical activity with less severe parkinsonism was not due to physical activity modifying the association of ADRD pathologies with the severity of parkinsonism.

## Discussion

Using novel data from almost 450 community-dwelling older participants, a higher level of total daily physical activity proximate to death was associated with less severe parkinsonism. Postmortem indices of ADRD pathologies did not mediate the association of physical activity with parkinsonism. Rather, we found that both physical activity and ADRD pathologies were independently associated with the level of parkinsonism proximate to death. These findings were robust and did not change when controlling for a wide range of clinical and health covariates. Further analyses did not show evidence for an interaction between physical activity and brain pathology in their association with the severity of parkinsonism, and hence, the interaction does not underlie the motor benefits of physical activity. Together, these findings highlight the need for further studies to elucidate the biology underlying the motor benefits of a more active lifestyle in older adults.

While the health benefits of higher levels of physical activity are well-known, the mechanisms underlying its potential benefits to reduce the severity of parkinsonism is unknown. Prior mouse studies suggest that higher levels of physical activity were associated with less amyloid and phosphorylated tau deposition, higher levels of markers of neurogenesis, and better spatial memory [26,27]. However, due to limited brain tissues available in well-characterized older adults, few studies have examined the association of physical activity and ADRD pathologies in older adults, and extant amyloid brain imaging studies have shown conflicting results [28].

Our prior studies have shown that a higher level of physical activity at study baseline in this cohort is associated with a slower progression of parkinsonism as well as a decreased incident of parkinsonism during an average of four years of follow-up. Additional studies have shown indices of brain pathology are associated with lower levels of total daily physical activity and more severe parkinsonism [9,10]. Together, these studies provided the scientific framework of the current study suggesting the possibility that brain pathologies may link (mediate) the association of higher levels of physical activity with less severe parkinsonism.

The current study replicated previous findings showing that a higher level of physical activity is associated with less severe parkinsonism [6]. While this association was significant, total daily physical activity explained only 7.2% of the variance of parkinsonism. This finding is similar with a previous study which reported that total daily physical activity was mildly correlated with a different motor phenotype which summarized several motor abilities, including gait speed, grip strength, and finger tapping and explained 9% of its variance [29].

Total daily physical activity and motor phenotypes may be weakly correlated as they assess different facets of motor function, a complex multidimensional phenotype. It has been suggested that total daily physical activity measures the quantity of volitional activity that occurs during the day. In contrast, other motor phenotypes may assess innate motor abilities. A person with poor motor abilities might nonetheless be very active and, conversely, an individual with good motor abilities might voluntarily elect not to move.

Similar to prior studies, in the current study brain pathologies were related to parkinsonism, but only accounted for a minority of its variance [30]. The accumulation of ADRD

pathologies in motor regions outside the brain that were not measured in this study [31] may account for the small amount of the variance of parkinsonism explained by brain ADRD pathologies. In addition, while the amount of parkinsonism variance explained by the physical activity or ADRD pathologies is small at the individual level their effect at the public health level is not inconsequential. Given the extent of motor impairment in old age, even the modest effect sizes observed in the current study are likely to be very important.

The current analyses did not find evidence that ADRD pathologies link physical activity with the severity of parkinsonism. Rather, we found that both ADRD pathologies and physical activity were independently associated with parkinsonism. Recent studies have suggested that some clinical covariates may modify indices of brain pathology and mitigate its untoward clinical effects [22–25]. The current study did not find evidence that the motor benefits of physical activity are due the interaction and modification of indices of ADRD brain pathologies with the severity of parkinsonism in older adults.

The current results with parkinsonism are similar to a prior study, in this cohort, in which we found that a higher level of total daily physical activity in older adults was associated with better cognition and this association was also independent of the presence of brain ADRD pathologies [29]. Thus, both the potential cognitive and motor benefits of a higher level of physical activity in older adults were unrelated to the presence of indices of brain pathologies, and there was also no evidence that physical activity modify the association of brain pathologies with parkinsonism or cognition.

As discussed below, these cross-sectional findings derive from observational data and conclusive causal inferences require further interventional studies. Yet, the current study's findings together with prior longitudinal studies which show that higher levels of baseline physical activity are associated with slower progressive parkinsonism[6] and cognitive decline[14] may lend support for interim public health strategies until more conclusive data is available. Currently, there are no treatments for AD and other degenerative brain pathologies. Findings that the potential clinical benefits of physical activity are unrelated to indices of brain pathology lend support public health efforts to increase the level of physical activity as a means to maintain both physical and cognitive function in older adults, even in the absence of efficacious treatments for ADRD pathologies.

If the results of the current study are replicated what potential mechanisms might underlie the motor benefits of physical activity? Most prior studies about the beneficial effects of life style factors have examined their association with cognition, not motor function. Like physical activity, varied factors such as education[32,33], late-life cognitive[34] and physical activities [29], and psychological factors like purpose in life[23] and social networks[35] were associated with slower cognitive decline independent of AD pathology. These data like the current data highlight that physical activity and other lifestyle factors may be linked with cognition via unidentified mechanisms which lack a known "pathologic footprint". Results of some human brain imaging studies suggest that increased physical activity may lead to better memory through vascular and structural brain changes [36]. Recently, system biology and network-based approaches have been exploited to leverage transcriptome data to identify genes and proteins i.e. molecular brain mechanisms that may drive cognition [37]. For example, higher levels of inositol 1, 3, 4-triphosphate 5/6-kinase (ITPK1) was associated with slower cognitive decline, but its association with cognition was independent of the presence of ADRD pathologies [38]. A similar approach may be useful to elucidate novel molecular mechanisms that underlie the potential motor and cognitive benefits of a more active lifestyle in older adults.

The biology underlying the association of life style factors with motor function is unclear. Two prior studies have suggested that education and physical activity modify the association of white matter hyperintensities with motor function [24,39]. These studies examined brain

imaging indices in living adults, but did not examine indices of ADRD pathologies. While the current study found that physical activity did not modify the association of ADRD with motor function, this study did not examine brain imaging indices. Therefore, further studies will need to examine brain imaging together with ADRD pathologies indices. In examining the underlying mechanisms of the association between life style factors and motor function, it is also important to consider the widespread distribution of motor-related CNS regions. In contrast to cognition, the motor system extends beyond the brain to influence spinal cord structures which via the peripheral nervous system regulate muscle structure, the final effector of all movement. Damage to any portion of this distributed motor system can impair motor function. Recent work has shown that indices of ADRD extend beyond the brain and brainstem to reach the spinal cord and are related to the severity of parkinsonism in older adults [31]. Further work is needed to extend the collection of indices of ADRD pathologies in postmortem studies to outside of the brain and also include brainstem and spinal cord motor structures. Finally, no prior work has examined the role of motor unit structures (spinal motor neuron, peripheral nerve and muscle) in linking ADRD phenotypes with parkinsonism in older adults [40]. These motor tissues will also be need to be examined to identify genes and proteins that may underlie the motor benefits of physical activity without manifesting a pathologic footprint.

Another question raised by the current study is the extent to which the association of higher levels of physical activity with better cognition and motor function derive from similar or distinct central nervous system (CNS) loci or mechanisms. For example recent work suggests that physical activity may cause muscle to release a hormone-like factor, irisin, that may reach cognitive brain regions via systemic circulation to protect cognition[40,41]. Finally, recent work suggests that molecular brain mechanisms without a known "pathologic" footprint may drive both cognitive and motor function[38,42].

Our study has several strengths. We analyzed quantitative metrics of physical activity derived from multiday continuous recordings obtained during everyday living in the community-setting. Structured and validated methods were employed to assess parkinsonism and brain autopsies. The combination of antemortem measures of physical activity and parkinsonism with neuropathological indices only available postmortem is especially challenging from a logistics perspective and adds to the strength of this work.

Our study has important limitations. The study was composed of highly educated Caucasians limiting generalizability of the study findings. However, the frequency of ADRD pathologies is this cohort is similar to other clinical-pathological longitudinal studies of aging (S6 Table) [43]. Another limitation is the cross sectional design of this study limiting causal inferences, as we cannot rule out reverse causality i.e., that reduced parkinsonism might lead to reduced physical activity. However, a prior longitudinal study in this same cohort showed that higher baseline total daily physical activity was associated with slower rate of progressive parkinsonism over time lending support for the potential beneficial effect of physical activity on the severity of parkinsonism [6]. Future longitudinal physical activity interventional studies employing repeated imaging of brain pathologies together with clinical data might be needed and might identify links between physical activity and parkinsonism not observed in the current study.

Moreover, prior data about how active the individuals in the current study were over the course of their lifespan was not available. Therefore, it is unclear if the benefit of a more active lifestyle is due to lifestyle choices over many years or due to choices during this study. Brain imaging indices were not examined in these analyses. Since the motor system extends beyond the brain, further work is needed to determine if ADRD pathologies in motor tissues outside the brain might mediate the association of total daily physical activity with the severity of

parkinsonism [41,44]. Finally, the activity monitors employed in this study do not differentiate between different physical activities, and further studies are needed to determine whether the specific types of movements associated with less severe parkinsonism. In addition, removal of the device employed in the current study cannot always be distinguished from periods of no activity.

Despite its limitations, accumulating more than 450 autopsies from well-characterized older adults with both quantitative multi-day activity metrics as well as assessments of parkinsonism proximate to death provides novel data which has potential to inform on the design of both clinical and interventional studies of physical activity in older adults. Further studies are needed to elucidate the knowledge gaps about the mechanisms underlying the benefits of physical activity. This is crucial to facilitate interventions that modify lifestyles as a means to maintain physical and cognitive function in our aging population.

## Supporting information

**S1 Table. Comparison of participants included and excluded from these analyses due to missing clinical data.**
(DOCX)

**S2 Table. Association of total daily physical activity proximate to death with postmortem brain pathology indices**∗**.**
(DOCX)

**S3 Table. Association of total daily physical activity and indices of brain pathologies with parkinsonism proximate to death controlling for potential confounders**∗**.**
(DOCX)

**S4 Table. Association of total daily physical activity and indices of brain pathologies with parkinsonism proximate to death after exclusion of participants with a clinical diagnosis of Parkinson Disease (PD) or those receiving neuroleptic medications.**
(DOCX)

**S5 Table. Association of total daily physical activity and indices of brain pathologies with parkinsonism proximate to death controlling for $k_{RA}$, probability metric of sleep disruption due to movement)**∗**.**
(DOCX)

**S6 Table. Distribution of indices of Alzheimer's Disease and Related Disorders (ADRD) pathologies in the Memory and Aging Project compared to 2 other community-based longitudinal clinical-pathological studies of aging(1).**
(DOCX)

## Acknowledgments

We thank participants of the Rush Memory and Aging project. Also, we appreciate staff of the Rush Alzheimer's Disease Center.

## Author Contributions

**Conceptualization:** Shahram Oveisgharan, Aron S. Buchman.

**Data curation:** Aron S. Buchman.

**Formal analysis:** Shahram Oveisgharan, Sue E. Leurgans, Lei Yu.

**Funding acquisition:** David A. Bennett, Aron S. Buchman.

**Investigation:** Julie A. Schneider, David A. Bennett, Aron S. Buchman.

**Methodology:** Shahram Oveisgharan, Robert J. Dawe, Sue E. Leurgans, Lei Yu, Julie A. Schneider, David A. Bennett, Aron S. Buchman.

**Project administration:** Julie A. Schneider, David A. Bennett.

**Resources:** David A. Bennett.

**Supervision:** Aron S. Buchman.

**Validation:** Aron S. Buchman.

**Visualization:** Aron S. Buchman.

**Writing – original draft:** Shahram Oveisgharan, Aron S. Buchman.

**Writing – review & editing:** Shahram Oveisgharan, Robert J. Dawe, Sue E. Leurgans, Lei Yu, Julie A. Schneider, David A. Bennett, Aron S. Buchman.

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
