## [Decision Letter · Decision Letter 0]

19 Feb 2020

PONE-D-19-27901

Total daily physical activity, brain pathologies, and parkinsonism in older adults

PLOS ONE

Dear Dr. Ovelsgharan,

Thank you for submitting your manuscript to PLOS ONE. We apologize that review of this manuscript took longer than usual. After careful consideration, we feel that it has merit but does not fully meet PLOS ONE’s publication criteria as it currently stands. Therefore, we invite you to submit a revised version of the manuscript that addresses the points raised during the review process.

The reviewer comments are provided below, and the following are additional editor comments:

There are a number of concerns related to the statistical analyses that require careful revision in the re-submission:The introduction is well-written, but does not support the methods or results sections as presented in the abstract or the manuscript. The introduction implies that a mediation analysis was conducted to test the mediating effect of brain pathologies on the relationship between PA and parkinsonism. However, the statistical analyses appear to test the association between PA and parkinsonism *independent *of brain pathologies and to test the interaction between PA and brain pathologies. These analyses do not reflect the research question as stated in the introduction section.The number of analyses is unfocused and raises concerns whether hypotheses were developed a priori.It appears that the four parkinsonism signs were tested in separate regression models. This is problematic, as the four subscales are likely correlated.As Reviewer 1 noted, the collinearity between brain pathology variables is not addressed. Were the physical activity data also positively skewed?While Figure 1 is pretty, it does not offer additional information not already available in Table 2.Please provide further information on the choice of covariates - it seems that only age at death and sex were included in the models, but other covariates (e.g., education, marital status, medications) may also be relevant.While the authors cite two previously published studies utilizing the physical activity data processing methods, it is not clear how sleep was accounted for in the data processing. The protocol utilizes 24-hour monitoring, and does not mention how sleep was filtered out. Movement during sleep time may represent disrupted sleep and should not be included in total activity counts. Please include a statement on how sleep time was handled in the data processing.Please pay specific attention to Review 1's comments related to the discussion section of the paper. Review 1 and I thought the presentation of the scientific evidence utilizing a logical argument of ABC was sufficient and helpful; however, Review 2 found it unclear. Please consider including a figure to illustrate the proposed scientific framework/analyses for the research question.Minor comments:Please include ranges in parentheses for parkinsonism variables in Table 1.

We would appreciate receiving your revised manuscript by March 23, 2020. To enhance the reproducibility of your results, we recommend that if applicable you deposit your laboratory protocols in protocols.io, where a protocol can be assigned its own identifier (DOI) such that it can be cited independently in the future. For instructions see: http://journals.plos.org/plosone/s/submission-guidelines#loc-laboratory-protocols

We look forward to receiving your revised manuscript.

Kind regards,

Diane K. Ehlers, PhD

Academic Editor

PLOS ONE

Journal Requirements:

Journal requirements met. Although Reviewer 2 felt that the data were not publicly available, the authors have adequately provided evidence of the availability of the data.

Reviewers' comments:

Reviewer's Responses to Questions

**Comments to the Author**

1. Is the manuscript technically sound, and do the data support the conclusions?

Reviewer #1: No

Reviewer #2: Yes

2. Has the statistical analysis been performed appropriately and rigorously? 

Reviewer #1: I Don't Know

Reviewer #2: Yes

3. Have the authors made all data underlying the findings in their manuscript fully available?

Reviewer #1: Yes

Reviewer #2: No

4. Is the manuscript presented in an intelligible fashion and written in standard English?

Reviewer #1: Yes

Reviewer #2: Yes

5. Review Comments to the Author

Reviewer #1: The manuscript leverages a rich autopsy dataset with associated clinical measures to test a mediation hypothesis that, brain pathology will mediate the relationship between physical activity (PA) and parkinsonism.

The introduction is clear, concise and well written. I especially appreciated the clear articulation of the mediation model and hypothesis. Thank you.

Some light editing is still required throughout for spelling and grammar.

I'm not sure I would say r of .4-.7 is "highly correlated". Might be better not to use an adjective here (p 7, line 134)

I think it would be important to note if there were any differences in PA (or other demographics) or other characteristics between those who had autopsy data, or UPDRS, and those who did not. I don't suspect there are but one never knows how selection bias might surface.

Help me understand the basis for your claim that the data support the idea that pathologies explain parkinsonism (Table 2, Model 2). 5.8% variance explained beyond age and sex seems a bit underwhelming (and there is no discussion of variance inflation amongst correlated neuropathologies, or presentation of the fully model, just the components). 7.8% variance explained by PA is a bit underwhelming too. However, I get that people are noisy.

I think my biggest concern in this manuscript is what seems to be the authors conclusion that the PAy measured can somehow be considered protective or "affording reserve". I know a veteran and established group such as this knows the inherent difficulty in identifying causality in regressions. Here one could easily interpret the data to mean that PAprotects against parkinsonism, or people with parkinsonism are less active. Thus the statement on p13,line 238 about physical activity affording reserve is ambitious at best. And the conclusion that PA in not interacting with brain pathology, when PA was measured so close to death is perhaps overly pessimistic for the field.

Again, on page 14, there is the hint of a conclusion about physical activity not affection neuropathology when the data provide no clear way to test this over time, or even with sufficient separation of activity data and neuropathology work up.

Given these concerns, the bulk of the discussion about reserve seems ambitious.

Reviewer #2: This manuscript sets out to describe the association of total daily physical activity and degree of parkinsonism in older adults near the end of life and to determine if that association is maintained when postmortem brain pathology indices are applied. The authors have previously established that higher levels of daily physical activity in older adults is associated with less severe parkinsonism. They have also previously shown that certain brain pathology indices related to more severe parkinsonism and in a separate publication that these brain pathology indices related to less total daily physical activities. The uniqueness of this current manuscript combines all three factors (totally daily physical activity, brain pathology, parkinsonism) in a single analysis in order to determine if higher level of physical activity relates to lower degree of parkinsonism and that this relationship is not diminished by the addition of brain pathology indices into the analysis. This analysis provides unique data to the field of exercise in the aging population and is in proper context of previous publications.

The analysis includes data from 447 subjects who participated in a community-based brain bank. The degree of parkinsonism is derived from a modified United Parkinson’s Disease Rating Scale (UPDRS) that has previously been verified. Total daily activity measures come from an accelerometer worn by participants which has also been previously verified. Brain pathology information obtained from blinded pathologist who evaluated for degree of Alzheimer’s disease and related disorders (ADRD) pathologies. A series of multivariable age and sex-adjusted linear regressions were used to assess the whether the association of daily physical activity with parkinsonism persisted when indices of brain pathology were included. The results of these analyses show higher levels of daily activity were associated with less severe parkinsonism; higher burden of brain pathology was associated with more severe parkinsonism; and that the association between increased physical activity resulting in less parkinsonism was not reduced by presence of brain pathology. The data they present do support the claims of the authors and the summary of current data in the field nicely demonstrates the importance of this current work. I do have the following minor recommendations to improve upon the clarity and quality of the publication, presented in the order they occurred in the manuscript:

#1: In the introduction (page 3; starting line 57) the terms “physical activity” “parkinsonism” and “brain pathology” are given A, C and B labeling but I found this confusing to follow while reading the text and did not aid in the understanding of the framework. Initially I thought this was going to relate into an equation but this labeling was not mentioned again and thus would recommend removing it.

#2: Page 4; Methods sections: I think it is pertinent to explicitly include if the cohort from the MAP is a “aging healthy control” database, or if these participants were recruited in the brain bank because they had cognitive decline or parkinsonism. This will help to provide further insight in how to interpret the data.

#3; Table 1 displays the clinical and postmortem characteristics of the cohort. It may be useful to add in this section that these characteristics (if indeed true) are similar to other reported clinical and brain bank findings in similarly aged subjects. This would serve to boost the generalizability of the data provided. For example – does the literature report similar findings of predominately (69%) AD pathologic findings in an aging cohort? Likewise is the degree of global parkinsonism similar to those also reported? Since this journal will attract more of a general audience; I think this will be helpful to put the results into perspective to those readers that are not as intricately aware of this data.

#4; Table 2: Would consider re-naming the Models with a more descriptive title to increase the clarity of what the linear regression models are representing.

#5: Table 2: in the final row: “Additional Explained Variance” are these values shown without the 2.5% addition of the variance due to age/sex as referenced in the caption? I am nearly certain that is what is trying to be related, but the language in the caption and Table 2 could be more clear. Perhaps, “an additional 2.5% of the variance is due to age/sex” or something similar could replace the “compared to age and sex” language.

#6; Page 13, Line 223: Regarding the statement (higher burden of brain pathology related to more severe parkinsonism) made about Model 2 – I am unclear about which data in Table 2 represent this statement. There does not appear to be a “global pathology burden” index to represent the overall burden of pathology; instead each individual pathology is listed and its individual relationship to parkinsonism. Additionally the linear regression models are not significant as P values are greater than 0.05 except for Nigral neuronal loss and Atherosclerosis. The association between these 2 pathologies and parkinsonism is mentioned in the text but I feel that the statement “higher burden of brain pathologies was associated with more severe parkinsonism” is unfounded because the bulk of the brain pathology did not show this same association.

#7: Page 13, line 229: Then sentence starting “In the latter model…” It is not clear which model you are referring to and again would be more clear if the models were renamed or at least directly referenced in this sentence. Is this referring to Model 2? If so, the variance mentioned in the text of 5.7% does not match the corresponding variance in Table 2 of 5.8%.

#8: Figure 1 section: The text starting on 233 and ending on 237: “The blue regression line….pathologies to the first model” seem more appropriate for the Figure 1 caption.

#9 Figure 1 section: Two points – 1) line 235 “A second red line..” are there supposed to be two red lines or should this be worded “The red line…” ? 2) The red regression line is displaying Model 3 from the description provided but the text references Models 1 and 2 at the start of this section. It is not clear to me which model this red line is representing- I think it is Model 3 but this should be more clearly stated.

#9: Figure 1: Overall this figure doesn’t really add to any new information except for a different way to represent the data from Table 2. I would recommend cutting this figure all together. If the goal is present a more “graphic” form of the information (which in my opinion is more digestible) – then the opposite approach could be considered and would suggest creating figures to represent the other models as well. If this approach was done, then Table 2 could be made available as a supplement table so as not to give redundant data in the body of the text.

#10: Page 15, line 267-268: The claim that higher level of daily activity is associated with lower odds of tremor does not appear to be true as the p value is 0.066

#11: Table 3: It is not clear what is meant by predictors “with TDPA” and “without TDPA.” It seems like these rows are representing data with and without the inclusion of the brain indices and perhaps better labeling would add clarity. Further explanation is needed to explain this figure and the significance it has in terms of the rest of the data presented.

#12: The PLOS One journal states that all data underlying the findings should available. There are 3 instances where the authors state that “data is not shown.” These occur in Line 248 as well as twice in Table 3. I would recommend sharing this data as a supplement.

6. PLOS authors have the option to publish the peer review history of their article (what does this mean?). If published, this will include your full peer review and any attached files.

Reviewer #1: No

Reviewer #2: No

---

## [Author Response · Author response to Decision Letter 0]

24 Mar 2020

Dear Dr. Ehlers

Thank you for the opportunity to revise and resubmit our manuscript. We thank you and the reviewer’s for the careful reading and helpful comments. A point by point response to each comment is included below. Material changes to the manuscript are noted below and changes are highlighted in the revised manuscript.

Kind regards,

Shahram Oveisgharan, MD

EDITORS’ COMMENTS:

E.1 The introduction is well-written, but does not support the methods or results sections as presented in the abstract or the manuscript. The introduction implies that a mediation analysis was conducted to test the mediating effect of brain pathologies on the relationship between PA and parkinsonism. However, the statistical analyses appear to test the association between PA and parkinsonism independent of brain pathologies and to test the interaction between PA and brain pathologies.

We hope that the changes to the revised the manuscript align the introduction, methods and results to support the proposed mediation hypothesis tested in our primary analysis (Introduction, (P3-4, lines 53-73), Methods (P10-11, lines 217-231), and Results (P14-17, lines 263-280)). As suggested by the reviewer as well as the editor (E.10), we have added a figure that summarizes and illustrates the scientific framework and hypothesis tested in this manuscript (Figure 1). 

E.2 The number of analyses is unfocused and raises concerns whether hypotheses were developed a priori.

We hope that the scientific framework outlined in the Introduction (Page 3-4, lines 53-73) and illustrated in Figure 1 underscore that the hypothesis tested was developed a priori and that changes in the main text focus the manuscript on the proposed mediation hypothesis. To further focus the paper, we have reduced the number of secondary analyses by removing the analysis of parkinsonian signs. The goal of our primary analysis was to elucidate whether mediation is a potential mechanism underlying the motor benefits of higher levels of physical activity. Multiple pathways likely underlie the health benefits of physical activity. To increase the scientific impact of this manuscript and maximally leverage the unique data used in these analyses, we retained the secondary analysis, during which a second potential mechanism distinct from mediation analysis was examined. In this secondary analysis, we examined whether the motor benefits of physical activity was via effect-modification of pathologies underlying parkinsonism (P18-19, lines 310-319) 

E.3 It appears that the four parkinsonism signs were tested in separate regression models. This is problematic, as the four subscales are likely correlated.

We agree with the editor that these signs are related and this is why in our earlier publications, supported by principal component analysis (PCA), these signs have been combined into the global parkinsonian score (1). Nonetheless, as additional analyses were requested (E5, E7) and to maintain the focus of this manuscript (E2), we have removed the analysis of the four individual parkinsonian signs from the manuscript. 

E.4 As Reviewer 1 noted, the collinearity between brain pathology variables is not addressed.

We examined collinearity among the brain pathology indices by calculating their variance inflation factor (VIF) (Table 2, model 2). The VIfs were less than 1.5 indicating that collinearity did not exist among the examined pathology indices(2). We added relevant sentences to the text indicating lack of collinearity among the examined brain pathologies (P11, lines 231-233; P17, lines 275-276). 

E.5 Were the physical activity data also positively skewed?

The skewness of the total daily physical activity was 1.3 and of the intensity of daily activity was 0.9. As physical activity metrics were predictors, not outcome variables, in our study that had a sample size of 447 providing reliable mean and standard deviations for our variables (3) we did not use the square-root transformation of the physical activity data. Nonetheless, we repeated our analyses after using square-root transformation of both metrics of physical activity (total daily physical activity and intensity of the physical activity). Using their square root transformation did not change the conclusion that a higher level of physical activity was associated with less severe parkinsonism. In linear regressions controlled for age at death and sex, a higher level of square root of total daily physical activity (estimate = -0.868, SE = 0.123, p<0.001) or square root of intensity of physical activity (estimate = -4.183, SE = 0.567, p<0.001) were associated with less severe parkinsonism proximate to death. 

E.6 While Figure 1 is pretty, it does not offer additional information not already available in Table 2.

We removed the previous Figure 1 and replaced it with the Figure suggested by the editor in comment E.1.

E.7 Please provide further information on the choice of covariates - it seems that only age at death and sex were included in the models, but other covariates (e.g., education, marital status, medications) may also be relevant.

Additional analyses have been added adjusting our models for education, marital status, vascular risk factors and diseases, and use of neuroleptic medications and our results were unchanged. These analyses are summarized in the main text [methods (P11, lines 233-235), results (P17, lines 284-289), Table 1] and included in a supplementary table (Supplementary Table e-3). 

E.8 While the authors cite two previously published studies utilizing the physical activity data processing methods, it is not clear how sleep was accounted for in the data processing. The protocol utilizes 24-hour monitoring, and does not mention how sleep was filtered out. Movement during sleep time may represent disrupted sleep and should not be included in total activity counts. Please include a statement on how sleep time was handled in the data processing.

We agree that some movements during sleep may represent disrupted sleep and we have developed a metric to capture and quantify movements related to periods of sleep/rest. Conventional sleep diaries are not employed in MAP because of participant’s burden, difficulty in minimizing missing data, and progressive cognitive impairment that affects the validity and reliability of these data. Moreover, in this cohort sleep or naps are not restricted to the night. On average 60% of the 24 hour recording shows no activity (either naps or sleep). Thus, periods of rest or sleep are not restricted to the “night”, but are present throughout the 24 recording making it impractical to filter out movement during “rest” or “sleep”. 

To address the editor’s concern, we used a previously validated metric that quantifies the probability of rest/sleep disruption by movement, and is derived from the continuous 24 hour actigraphic recordings. KRA is the transition probability once sustained rest is achieved. This metric examines the temporal organization of human rest-activity patterns in terms of transition probabilities between periods of rest and activity. Conceptually, kRA is a measure of the tendency for disruption of rest (no activity) by physical activity. A higher value of kRA represents more fragmented sleep/rest - more movement during periods of sleep/rest. A lower value represents a more consolidated sleep/rest. Therefore, kRA indirectly captures movement/physical activity during sleep. KRA was weakly correlated with total daily physical activity (Spearman correlation coefficient = 0.15, p = 0.003). 

To adjust for sleep-related movement, we added a term for kRA to our primary model and the association of total daily physical activity with parkinsonism in presence of brain pathology indices was unchanged. These data are summarized in the Results (P18, lines 297-307), and the full models are included in the Supplementary Table e-5. Kra is described in the methods (P6-7, lines 124-135), and has been added to Table 1. 

E.9 Please pay specific attention to Review 1's comments related to the discussion section of the paper.

We hope that the changes to the discussion described below in response to Reviewer 1’s comments have adequately addressed these concerns. 

E.10 Review 1 and I thought the presentation of the scientific evidence utilizing a logical argument of ABC was sufficient and helpful; however, Review 2 found it unclear. Please consider including a figure to illustrate the proposed scientific framework/analyses for the research question.

We added a figure, Figure 1, illustrating the proposed scientific framework and proposed hypothesis testing in this manuscript.

E.11 Please include ranges in parentheses for parkinsonism variables in Table 1.

We included ranges in parentheses for the parkinsonism variables in Table 1. 

REVIEWER 1 

R1.1 Some light editing is still required throughout for spelling and grammar.

Thank you for pointing this out and we hope we have corrected all spelling and grammar errors. 

R1.2 I'm not sure I would say r of .4-.7 is "highly correlated". Might be better not to use an adjective here (p 7, line 134).

We deleted “highly” in the mentioned sentences. 

R1.3 I think it would be important to note if there were any differences in PA (or other demographics) or other characteristics between those who had autopsy data, or UPDRS, and those who did not. I don't suspect there are but one never knows how selection bias might surface.

These data have been added to the Supplementary Table e-1 comparing included and excluded participants in the demographics and clinical variables and summarized in the main text (P5, lines 92-96).

R1.4 Help me understand the basis for your claim that the data support the idea that pathologies explain parkinsonism (Table 2, Model 2). 5.8% variance explained beyond age and sex seems a bit underwhelming (and there is no discussion of variance inflation amongst correlated neuropathologies, or presentation of the fully model, just the components).

We agree with the reviewer. While this manuscript is focused on physical activity, our prior publications have focused on this important question raised by the reviewer (4–9). In prior reports which examining different motor phenotypes, we have found that indices of brain pathologies account for a only a small minority of the motor phenotype variance (5-15%)(9) much less than the 30-50% observed with cognitive phenoytpes (10). This question is the current focus of our research efforts that seek to determine the pathologic basis underlying progressive parkinsonism and motor decline in old adults. While networks that subserve cognition are found in the brain, the networks underlying motor function extend from the brain through the CNS to reach muscle in the periphery. Our recent work shows that ADRD pathologies extend beyond the brain to brainstem and spinal cord regions(11). Thus, to determine the full extent of the pathologic basis for parkinsonism as well as the mechanisms underlying the motor benefits of physical activity one needs to investigate the pathologies not only in the brain but also interrogate vital motor-related tissues outside the brain. This point is addressed in the Discussion, (P20-21, lines 359-363).

Moreover, while the amount of parkinsonism variance explained by the physical activity or ADRD pathologies is small at the individual level the effect at the public health level is not inconsequential. Given the extent of motor impairment in old age, that half of the adults 85 years or older have parkinsonian signs, even the modest effect sizes observed in the current study are important (12). These details are noted in the Discussion (P21, lines 363-366). 

Variance Inflation: We also examined collinearity among the brain pathology indices by calculating their variance inflation factor (VIF) (Table 2, model 2). The VIfs were less than 1.5 indicating that variance inflation did not occur among the examined pathology indices(2). These data have been added to the text (P10, lines 201-203; P14, lines 230-231).

R1.5 7.2% variance explained by PA is a bit underwhelming. However, I get that people are noisy.

Thank you for raising this important point. The variance accounted for by physical activity of parkinsonism is similar to that observed in a prior study(13) in which physical activity accounted for about 8% of the variance of a summary 10 motor performances. Some of the motor phenotypes assessed examine motor abilities. In contrast, total daily physical activity is a volitional behavior. Damage to any portion of the distributed motor system has potential to degrade motor abilities underlying movement while leaving motor decision-making regions necessary for the initiation of movement intact. Thus, an individual with poor motor abilities may nonetheless have higher levels of total daily physical activity compared to an individual with good motor abilities, who might elect to sit in a chair for the entire day. As illustrated in the scatterplot on the right(13), this may explain why the quantity of daily physical activity is only modestly related to motor abilities and not everyone with a high level of physical activity necessarily has good motor abilities. 

Finally, in prior publications most single risk factors significantly associated with motor phenotypes account for 2-3% of the variance of a motor phenotype (14). Therefore, at the population level, the fact that total daily physical activity accounts for 7% of the variance of parkinsonism suggests that it is an important risk factors for parkinsonism in older adults. We have added these points to the discussion (P20, lines 353-358; P21, lines 363-366). 

R1.6 I think my biggest concern in this manuscript is what seems to be the authors conclusion that the PAy measured can somehow be considered protective or "affording reserve". I know a veteran and established group such as this knows the inherent difficulty in identifying causality in regressions. Here one could easily interpret the data to mean that PAprotects against parkinsonism, or people with parkinsonism are less active. Thus the statement on p13, line 238 about physical activity affording reserve is ambitious at best. 

We agree that a well-known limitation of all cross-sectional observational studies is that causal inferences are limited due to the potential for reverse causality. This and other limitations affecting the current study are included in the discussion (P24, lines 443-451). However, as noted in the current discussion, we have published a longitudinal study in this same cohort which showed that a higher level of physical activity at study baseline was associated with a slower rate of progressive parkinsonism and decreased incidence of parkinsonism during four years of follow-up on average. In the absence of results from an interventional study supporting a causal relationship, our results are not conclusive. Yet, the results of the previous and current study lend support for considering the potential public health consequences of our current findings. Our findings inform on the need for further interventional studies of physical activity, and suggest potential utility of research efforts to identify molecular mechanisms without a pathologic footprint that drive the motor benefit of physical activity in older adults. The changes can be found at the following places in the manuscript (P21-22, lines 381-390; P24, lines 448-451). 

R1.7 And the conclusion that PA is not interacting with brain pathology, when PA was measured so close to death is perhaps overly pessimistic for the field. Again, on page 14, there is the hint of a conclusion about physical activity not affection neuropathology when the data provide no clear way to test this over time, or even with sufficient separation of activity data and neuropathology work up.

We agree with the reviewer that there are multiple limitations to our current study design. No single study is going to elucidate the complex biology underlying the benefits of physical activity in older adults. Despite the limitations noted by the reviewer, by using the same approach as the current study used we have found the following clinical covariates as modifiers of the relationship between postmortem pathology and cognition proximate to death: education, purpose in life, physical activity and postmortem BDNF gene expression levels derived from prefrontal cortex (15–18). Thus, in the absence of prior human data, despite limitations these observational cross sectional data provide novel data about an important potential mechanism that could account for the motor benefits of physical activity. We hope that the changes in the revised text highlights the limitations of this study in examining the interaction of total daily physical activity with ADRD pathologies in their association with parkinsonism (P12, lines 244-248; P19, lines 317-319; P21, lines 369-373), and address in part the concerns of the reviewer. 

REVIEWER 2

 R2.1 In the introduction (page 3; starting line 57) the terms “physical activity” “parkinsonism” and “brain pathology” are given A, C and B labeling but I found this confusing to follow while reading the text and did not aid in the understanding of the framework. Initially I thought this was going to relate into an equation but this labeling was not mentioned again and thus would recommend removing it.

Following the editor’s comment, E.1, we added a figure, Figure 1, illustrating scientific framework underlying current analyses. In addition, we revised the introduction to illustrate more clearly the underlying scientific framework of the study (P3, lines 53-59). 

R2.2 Page 4; Methods sections: I think it is pertinent to explicitly include if the cohort from the MAP is a “aging healthy control” database, or if these participants were recruited in the brain bank because they had cognitive decline or parkinsonism. This will help to provide further insight in how to interpret the data.

The study is a cohort of aging healthy community-dwelling participants, and we added this to the manuscript (P4, lines 76-77). 

R2.3 Table 1 displays the clinical and postmortem characteristics of the cohort. It may be useful to add in this section that these characteristics (if indeed true) are similar to other reported clinical and brain bank findings in similarly aged subjects. This would serve to boost the generalizability of the data provided. For example – does the literature report similar findings of predominately (69%) AD pathologic findings in an aging cohort? Likewise is the degree of global parkinsonism similar to those also reported? Since this journal will attract more of a general audience; I think this will be helpful to put the results into perspective to those readers that are not as intricately aware of this data.

We added the Supplementary Table e-6 including frequencies of ADRD pathology indices in our cohort in comparison to two other clinical-pathological studies. In addition, we added the limitations the current study has in the generalizability of its findings (P24, lines 440-443). 

R2.4 Table 2: Would consider re-naming the Models with a more descriptive title to increase the clarity of what the linear regression models are representing.

At Table 2, we added abbreviations to the models’ names indicating included predictors. The changes are: model 1 is changed to model 1-TDPA, model 2 to model 2-Path, model 3 to model 3-TDPA+Path, and model 4 to model 4-Intensity+path. In addition, TDPA and intensity are defined in the “Model Term” column of Table 2. 

R2.5 Table 2: in the final row: “Additional Explained Variance” are these values shown without the 2.5% addition of the variance due to age/sex as referenced in the caption? I am nearly certain that is what is trying to be related, but the language in the caption and Table 2 could be more clear. Perhaps, “an additional 2.5% of the variance is due to age/sex” or something similar could replace the “compared to age and sex” language.

We appreciate the suggestion. We added the suggested sentence to the footnote of Table 2, and deleted the “compared to age and sex” from the table.

R2.6 Page 13, Line 223: Regarding the statement (higher burden of brain pathology related to more severe parkinsonism) made about Model 2 – I am unclear about which data in Table 2 represent this statement. There does not appear to be a “global pathology burden” index to represent the overall burden of pathology; instead each individual pathology is listed and its individual relationship to parkinsonism. Additionally the linear regression models are not significant as P values are greater than 0.05 except for Nigral neuronal loss and Atherosclerosis. The association between these 2 pathologies and parkinsonism is mentioned in the text but I feel that the statement “higher burden of brain pathologies was associated with more severe parkinsonism” is unfounded because the bulk of the brain pathology did not show this same association.

We deleted the “higher burden of brain pathologies was associated with more severe parkinsonism” from the text. 

R2.7 Page 13, line 229: Then sentence starting “In the latter model…” It is not clear which model you are referring to and again would be more clear if the models were renamed or at least directly referenced in this sentence. Is this referring to Model 2? If so, the variance mentioned in the text of 5.7% does not match the corresponding variance in Table 2 of 5.8%.

We apologize. We named the model, which is model 3-TDPA-Path. In this model, 8.3% of the parkinsonism severity variance was explained by age, sex, and pathologies and an additional 5.7% by total daily physical activity. 

R2.8 Figure 1 section: The text starting on 233 and ending on 237: “The blue regression line….pathologies to the first model” seem more appropriate for the Figure 1 caption.

We deleted Figure 1 and its section in the text, following E.6 and R2.9. 

R2.9 Figure 1 section: Two points – 1) line 235 “A second red line..” are there supposed to be two red lines or should this be worded “The red line…” ? 2) The red regression line is displaying Model 3 from the description provided but the text references Models 1 and 2 at the start of this section. It is not clear to me which model this red line is representing- I think it is Model 3 but this should be more clearly stated. Figure 1: Overall this figure doesn’t really add to any new information except for a different way to represent the data from Table 2. I would recommend cutting this figure all together. If the goal is present a more “graphic” form of the information (which in my opinion is more digestible) – then the opposite approach could be considered and would suggest creating figures to represent the other models as well. If this approach was done, then Table 2 could be made available as a supplement table so as not to give redundant data in the body of the text.

We deleted Figure 1 and its section in the text, following E.6 and R2.9.

R2.10 Page 15, line 267-268: The claim that higher level of daily activity is associated with lower odds of tremor does not appear to be true as the p value is 0.066

Following E.2 comment, we deleted Table 3 and related segment from the paper to make the analyses more focused, and the paper clearer. 

R2.11 Table 3: It is not clear what is meant by predictors “with TDPA” and “without TDPA.” It seems like these rows are representing data with and without the inclusion of the brain indices and perhaps better labeling would add clarity. Further explanation is needed to explain this figure and the significance it has in terms of the rest of the data presented.

Following E.2 comment, we deleted Table 3 and related segment from the paper to make the analyses more focused, and the manuscript clearer.

R2.12 The PLOS One journal states that all data underlying the findings should available. There are 3 instances where the authors state that “data is not shown.” These occur in Line 248 as well as twice in Table 3. I would recommend sharing this data as a supplement.

This data has been added as Supplementary Table e-4. Table 3 are deleted following E.2 comments. 

REFERENCES

1. Bennett DA, Shannon KM, Beckett LA, Wilson RS. Dimensionality of parkinsonian signs in aging and Alzheimer’s disease. J Gerontol Biol Sci Med Sci. 1999 Apr;54(4):M191-6. 

2. Dormann CF, Elith J, Bacher S, Buchmann C, Carl G, Carré G, et al. Collinearity: a review of methods to deal with it and a simulation study evaluating their performance. Ecography. 2013 Jan;36(1):27–46. 

3. Piovesana A, Senior G. How Small Is Big: Sample Size and Skewness. Assessment. 2018 Sep;25(6):793–800. 

4. Buchman AS, Leurgans SE, Nag S, Bennett DA, Schneider JA. Cerebrovascular disease pathology and parkinsonian signs in old age. Stroke. 2011 Nov;42(11):3183–9. 

5. Buchman AS, Shulman JM, Nag S, Leurgans SE, Arnold SE, Morris MC, et al. Nigral pathology and parkinsonian signs in elders without Parkinson disease. Ann Neurol. 2012 Feb;71(2):258–66. 

6. Buchman AS, Yu L, Boyle PA, Levine SR, Nag S, Schneider JA, et al. Microvascular brain pathology and late-life motor impairment. Neurology. 2013 Feb 19;80(8):712–8. 

7. Buchman AS, Yu L, Wilson RS, Boyle PA, Schneider JA, Bennett DA. Brain pathology contributes to simultaneous change in physical frailty and cognition in old age. J Gerontol Biol Sci Med Sci. 2014 Dec;69(12):1536–44. 

8. Buchman AS, Nag S, Shulman JM, Lim ASP, VanderHorst VGJM, Leurgans SE, et al. Locus coeruleus neuron density and parkinsonism in older adults without Parkinson’s disease. Mov Disord. 2012 Nov;27(13):1625–31. 

9. Buchman AS, Yu L, Wilson RS, Leurgans SE, Nag S, Shulman JM, et al. Progressive parkinsonism in older adults is related to the burden of mixed brain pathologies. Neurology. 2019 Apr 16;92(16):e1821–30. 

10. Boyle PA, Wilson RS, Yu L, Barr AM, Honer WG, Schneider JA, et al. Much of late life cognitive decline is not due to common neurodegenerative pathologies: Cognitive Decline. Ann Neurol. 2013 Sep;74(3):478–89. 

11. Buchman AS, Leurgans SE, Nag S, VanderHorst V, Kapasi A, Schneider JA, et al. Spinal Arteriolosclerosis Is Common in Older Adults and Associated With Parkinsonism. Stroke. 2017 Oct;48(10):2792–8. 

12. Louis ED, Luchsinger JA, Tang MX, Mayeux R. Parkinsonian signs in older people: prevalence and associations with smoking and coffee. Neurology. 2003 Jul 8;61(1):24–8. 

13. Buchman AS, Yu L, Wilson RS, Lim A, Dawe RJ, Gaiteri C, et al. Physical activity, common brain pathologies, and cognition in community-dwelling older adults. Neurology. 2019 Feb 19;92(8):e811–22. 

14. Oveisgharan S, Yu L, Dawe RJ, Bennett DA, Buchman AS. Total daily physical activity and the risk of parkinsonism in community-dwelling older adults. J Gerontol Biol Sci Med Sci. 2019; 

15. Bennett DA, Wilson RS, Schneider JA, Evans DA, Mendes de Leon CF, Arnold SE, et al. Education modifies the relation of AD pathology to level of cognitive function in older persons. Neurology. 2003 Jun 24;60(12):1909–15. 

16. Buchman AS, Yu L, Boyle PA, Schneider JA, De Jager PL, Bennett DA. Higher brain BDNF gene expression is associated with slower cognitive decline in older adults. Neurology. 2016 Feb 23;86(8):735–41. 

17. Boyle PA, Buchman AS, Boyle PA, Yu L, Schneider JA, Bennett DA. Effect of Purpose in Life on the Relation Between Alzheimer Disease Pathologic Changes on Cognitive Function in Advanced Age. Arch Gen Psychiatry. 2012 May 1;69(5):499. 

18. Fleischman DA, Yang J, Arfanakis K, Arvanitakis Z, Leurgans SE, Turner AD, et al. Physical activity, motor function, and white matter hyperintensity burden in healthy older adults. Neurology. 2015 Mar 31;84(13):1294–300.

---

## [Decision Letter · Decision Letter 1]

15 Apr 2020

Total daily physical activity, brain pathologies, and parkinsonism in older adults

PONE-D-19-27901R1

Dear Dr. Oveisgharan,

We are pleased to inform you that your manuscript has been judged scientifically suitable for publication and will be formally accepted for publication once it complies with all outstanding technical requirements.

With kind regards,

Diane K. Ehlers, PhD

Academic Editor

PLOS ONE

Additional Editor Comments (optional):

Please review for minor grammatical errors prior to production.

Reviewers' comments:

Reviewer's Responses to Questions

**Comments to the Author**

1. If the authors have adequately addressed your comments raised in a previous round of review and you feel that this manuscript is now acceptable for publication, you may indicate that here to bypass the “Comments to the Author” section, enter your conflict of interest statement in the “Confidential to Editor” section, and submit your "Accept" recommendation.

Reviewer #1: All comments have been addressed

Reviewer #2: All comments have been addressed

2. Is the manuscript technically sound, and do the data support the conclusions?

Reviewer #1: Yes

Reviewer #2: Yes

3. Has the statistical analysis been performed appropriately and rigorously? 

Reviewer #1: Yes

Reviewer #2: Yes

4. Have the authors made all data underlying the findings in their manuscript fully available?

Reviewer #1: Yes

Reviewer #2: Yes

5. Is the manuscript presented in an intelligible fashion and written in standard English?

Reviewer #1: Yes

Reviewer #2: Yes

6. Review Comments to the Author

Reviewer #1: The authors have addressed my initial concerns. I have no further comments that I think would improve the readability or utility of the manuscript.

Reviewer #2: (No Response)

7. PLOS authors have the option to publish the peer review history of their article (what does this mean?). If published, this will include your full peer review and any attached files.

Reviewer #1: Yes: Eric Vidoni

Reviewer #2: Yes: Mara Seier

---

## [Editor Report · Acceptance letter]

20 Apr 2020

PONE-D-19-27901R1 

Total daily physical activity, brain pathologies, and parkinsonism in older adults 

Dear Dr. Oveisgharan:

I am pleased to inform you that your manuscript has been deemed suitable for publication in PLOS ONE. Congratulations! Your manuscript is now with our production department. 

With kind regards,

on behalf of

Dr. Diane K. Ehlers 

Academic Editor

PLOS ONE